# Interturn Short Fault Diagnosis Using Magnitude and Phase of Currents in Permanent Magnet Synchronous Machines

**DOI:** 10.3390/s22124597

**Published:** 2022-06-17

**Authors:** Hyeyun Jeong, Hojin Lee, Seongyun Kim, Sang Woo Kim

**Affiliations:** Department of Electrical Engineering, POSTECH, 77, Cheongam-Ro, Nam-Gu, Pohang 37673, Gyeongbuk, Korea; jhy90@postech.edu (H.J.); suvvus@postech.edu (H.L.); ksy3dmbe3kor@postech.edu (S.K.)

**Keywords:** permanent magnet synchronous machines, diagnosis, fault detection, interturn fault, fault current

## Abstract

With the increased demand for permanent magnet synchronous machines (PMSMs) in various industrial fields, interturn short fault (ITSF) diagnosis of PMSMs is under the limelight. In particular, to prevent accidents caused by PMSM malfunctions, it is difficult and greatly necessary to diagnose slight ITSF, which is a stage before the ITSF becomes severe. In this paper, we propose a novel fault indicator based on the magnitude and phase of the current. The proposed fault indicator was developed using analysis of positive-sequence current (PSC) and negative-sequence current (NSC), which means the degree of the asymmetry of the three-phase currents by ITSF. According to the analysis, as ITSF increases, the phase difference between PSC and NSC decreases and the magnitude of NSC increases. Therefore, the novel fault indicator is suggested as a product of the cosine value of the phase indicator and the magnitude indicator. The magnitude indicator is the magnitude of NSC, and the phase indicator means the phase difference between the PSC and the NSC. The suggested fault indicator diagnoses the degree of ITSF as well as slight ITSFs under various conditions by only measured three-phase currents. Experimental results demonstrate the effectiveness of our proposed method under various torque and speeds.

## 1. Introduction

Permanent magnet synchronous machines (PMSMs) are being increasingly used in diverse industries such as electronic appliances, electric vehicles, aviation, and military fields owing to their high power density, high efficiency, and low maintenance costs [1,2,3,4,5,6,7]. Faults in PMSMs can reduce use efficiency or even stop the overall system, leading to serious losses caused by accidents [8,9]. As such, simple and effective system monitoring and accurate fault diagnosis are essential to minimize asset losses and increase safety.

The interturn short fault (ITSF) of the stator is one of the most common faults in PMSMs [6,8,9]. ITSF causes fault currents that can generate excessive heat on short-circuit windings, and the degree of ITSF is described as the shorted turn ratio and fault resistance [10]. In general, an increase in the short turn ratio or a decrease in fault resistance results in a higher degree of ITSF. Moreover, ITSF is caused by voltage- and current-induced stress or heat-induced electric breakdown. A more serious electric breakdown of the windings causes it to be more easily distributed to more windings, thereby leading to massive losses. Therefore, it is greatly important to diagnose the slight ITSF before ITSF becomes serious.

Considerable research has been conducted to diagnose ITSF [10,11,12,13,14,15,16,17,18,19,20,21,22,23,24]. Methods to diagnose ITSF using only the electric current include frequency analysis methods such as fast Fourier transform, short-time Fourier transform, and wavelet transform, as well as second-harmonics analysis and symmetrical component analysis [10,11,12,13]. The diagnostic methods, which demand only the currents, are a significantly simple and powerful advantage, allowing them to be easily applicable to various industrial systems. However, these are limited in that they can only diagnose serious faults or need to know timing the faults started.

Next, ITSF can be diagnosed by data-based methods that use training data [14,15,16,17]. These methods utilize artificial neural networks, convolutional neural networks, and recurrent neural networks to diagnose faults. They have the advantage of diagnosing faults before the degree of ITSF become serious; however, the data-based methods require high volumes of fault data to be fed into the learning algorithm [3]. However, it is difficult for actual industrial systems to secure high volumes of fault data for PMSMs that have not experienced faults yet.

The last method is a model-based method using model formulae and parameters [18,19,20,21,22,23,24]. This method uses features such as the Kalman filter and zero-sequence components, etc., and it diagnoses faults based on the difference between the model and the measured value. However, to use this method, the operator must know motor parameter values such as resistance and inductance. Moreover, additional signal processing devices or measurement devices are required to sense various signals associated with the voltage and speed.

To overcome such limitations, we herein suggest a manner of diagnosing ITSFs using fault indicators from the magnitude and phase of the three-phase currents using positive-sequence current (PSC) and negative-sequence current (NSC) in PMSMs. First, an ITSF in a PMSM leads the three-phase current waveform to become asymmetric. As such, the focus is on PSC and NSC that expresses this state. Subsequently, the magnitude of NSC is used to define the magnitude indicator according to which faults increase as the fault becomes more serious. Then, the difference between the phases of PSC and NSC is used to define the phase indicator according to which faults decrease as the fault becomes more serious. The proposed ITSF indicator can effectively capture both the diagnosis of the slight ITSF and the degree of ITSFs. The superiority of the suggested fault index is proven by the experiment under various fault, torque, and speed conditions. In addition, as the suggested diagnosis method uses only the three-phase current, it does not require voltage and motor parameters. It can also diagnose ITSFs before they become serious, even under varying torque and speed conditions. In particular, this method is simple, and is useful in diagnosing ITSFs before they become serious in industrial sites or machine parameters have changed due to aging and faults, rendering it impossible to identify the parameter values.

This paper is organized as follows. Section 2 describes the ITSFs in PMSM and introduces the model formula for ITSFs. Section 3 defines the magnitude and phase indices of PSC and NSC and analyzes the characteristics depending on the seriousness of the fault. Section 4 discusses the experiment and its configuration. Section 5 presents and discusses the results. Finally, Section 6 provides concluding statements.

## 2. PMSM Dynamics with ITSFs

Figure 1 illustrates the winding configuration of a wye-connection with an ITSF in phase *a*. Here, $i_{a}$, $i_{b}$ and $i_{c}$ denote the currents of the phase *a*, *b* and *c*, respectively. $v_{a}$, $v_{b}$ and $v_{c}$ denote the voltages of the phase *a*, *b* and *c*, respectively.

Shorted turn-ratio μ defines the ratio of the number of short-circuited turns in each phase and the number of total turns in each phase. By definition, μ is zero in the normal state and is greater than zero in case of ITSF. The larger the μ, the greater the degree of ITSFs. The fault resistance $R_{f}$ defines the degree of insulation failure of the small closed loop formed by the ITSF shown in Figure 1. By definition, $R_{f}$ is infinite in the normal state and is finite value in case of ITSF. The smaller the $R_{f}$, the greater the degree of ITSFs. In summary, ITSF can be represented by μ and $R_{f}$, a normal state is indicated when μ is zero and $R_{f}$ is infinite, and a full ITSF is indicated when μ is 1 and $R_{f}$ is zero.

The stator equations of the PMSM with ITSF are as follows [19]:(1)$$\begin{array}{c} \left[v_{abc}\right]=R_{s}\left[i_{abc}\right]+\frac{d}{dt}\left(\left[L_{s}\right]\left[i_{abc}\right]+\left[\lambda_{PM,abc}\right]\right)-R_{s}\left[P\right]\mu i_{f}-\frac{d}{dt}\left(\left[L_{s}\right]\left[P\right]\mu i_{f}\right), \end{array}$$
where
$$\begin{array}{c} \left[v_{abc}\right]=\left[\begin{array}{c} v_{a}\\ v_{b}\\ v_{c} \end{array}\right],\;\left[i_{abc}\right]=\left[\begin{array}{c} i_{a}\\ i_{b}\\ i_{c} \end{array}\right],\;\left[L_{s}\right]=\left[\begin{array}{ccc} L & M & M\\ M & L & M\\ M & M & L \end{array}\right],\;\left[\lambda_{PM,abc}\right]=\left[\begin{array}{c} cos\theta \\ \;cos(\theta -\frac{2\pi }{3})\\ cos(\theta +\frac{2\pi }{3}) \end{array}\right]. \end{array}$$

$\left[v_{abc}\right]$ denotes the phase voltage matrix when ITSF occurs. $v_{a}$, $v_{b}$, and $v_{c}$ are the voltages of phase *a*, *b*, and *c* with ITSF in a PMSM, repectively. Similarly, $\left[i_{abc}\right]$ denotes the phase current matrix when ITSF occurs. $i_{a}$, $i_{b}$, and $i_{c}$ are the currents of phase *a*, *b*, and *c* with ITSF, respectively. $R_{s}$ is the stator resistance. $L_{s}$ is the stator inductance, *L* is the self-inductance, and *M* is the mutual inductance. $\left[\lambda_{PM,abc}\right]$ is the phase flux linkage matrix, $\lambda_{PM}$ is the flux linkage due to permanent magnet, and θ is the electrical angular position. $i_{f}$ is the fault current and [P] is the vector that determines a faulty phase. If an ITSF occurs in the phase *a*, *b*, or *c*, then $\left[P\right]={\left[\begin{array}{ccc} 1 & 0 & 0 \end{array}\right]}^{T}$, $\left[P\right]={\left[\begin{array}{ccc} 0 & 1 & 0 \end{array}\right]}^{T}$ or $\left[P\right]={\left[\begin{array}{ccc} 0 & 0 & 1 \end{array}\right]}^{T}$, respectively.

In addition, the fault voltage in the shorted turns is expressed as follows:(2)$$\begin{array}{c} v_{f}=R_{f}i_{f}=\mu {\left[P\right]}^{T}\left[v_{abc}\right]-\mu \left(1-\mu \right)R_{s}i_{f}. \end{array}$$

## 3. A Strategy for ITSF Diagnosis

### 3.1. Faulty Parts of the Three-Phase Currents

By the superposition principle, the stator currents can be expressed as a summation of healthy and faulty parts.
(3)$$\begin{array}{c} \left[i_{abc}\right]={\left[i_{abc}\right]}_{H}+{\left[i_{abc}\right]}_{F}, \end{array}$$
where:$$\begin{array}{c} {\left[i_{abc}\right]}_{H}=\left[\begin{array}{c} i_{aH}\\ i_{bH}\\ i_{cH} \end{array}\right],\;{\left[i_{abc}\right]}_{F}=\left[\begin{array}{c} i_{aF}\\ i_{bF}\\ i_{cF} \end{array}\right].\; \end{array}$$

${\left[i_{abc}\right]}_{H}$ denotes the phase current matrix of the healthy parts. $i_{aH}$, $i_{bH}$, and $i_{cH}$ are the currents of phase *a*, *b*, and *c* of the healthy parts, respectively. Similarly, ${\left[i_{abc}\right]}_{F}$ denotes the phase current matrix of the faulty parts. $i_{aF}$, $i_{bF}$, and $i_{cF}$ are the currents of phase *a*, *b*, and *c* of the faulty parts, respectively.

In the wye-connected winding, the summation of three-phase currents is zero:(4)$$\begin{array}{c} i_{a}+i_{b}+i_{c}=0, \end{array}$$
using (Equation 3), the healthy and faulty parts of (Equation 4) are expressed as follows, respectively:(5)$$\begin{array}{c} i_{aH}+i_{bH}+i_{cH}=0, \end{array}$$
(6)$$\begin{array}{c} i_{aF}+i_{bF}+i_{cF}=0. \end{array}$$

Meanwhile, using (Equation 3), the healthy and faulty parts of (Equation 1) are also expressed as follows, respectively:(7)$$\begin{array}{c} {\left[v_{abc}\right]}_{H}=R_{s}{\left[i_{abc}\right]}_{H}+\frac{d}{dt}\left(\left[L_{s}\right]{\left[i_{abc}\right]}_{H}+\left[\lambda_{PM,abc}\right]\right), \end{array}$$
(8)$$\begin{array}{c} {\left[v_{abc}\right]}_{F}=R_{s}{\left[i_{abc}\right]}_{F}+\frac{d}{dt}\left(\left[L_{s}\right]{\left[i_{abc}\right]}_{F}\right)-R_{s}\left[P\right]\mu i_{f}-\frac{d}{dt}\left(\left[L_{s}\right]\left[P\right]\mu i_{f}\right), \end{array}$$
where:$$\begin{array}{c} {\left[v_{abc}\right]}_{H}=\left[\begin{array}{c} v_{aH}\\ v_{bH}\\ v_{cH} \end{array}\right],\;{\left[v_{abc}\right]}_{F}=\left[\begin{array}{c} v_{aF}\\ v_{bF}\\ v_{cF} \end{array}\right],\; \end{array}$$

${\left[v_{abc}\right]}_{H}$ denotes the phase voltage matrix of the healthy parts. $v_{aH}$, $v_{bH}$, and $v_{cH}$ are the voltages of phase *a*, *b*, and *c* of the healthy parts, respectively. Similarly, ${\left[v_{abc}\right]}_{F}$ denotes the phase voltage matrix of the healthy parts. $v_{aF}$, $v_{bF}$, and $v_{cF}$ are the voltages of phase *a*, *b*, and *c* of the faulty parts, respectively.

For the convenience of understanding, assuming that ITSF occurred in phase *a* as shown in Figure 1, subtracting the third row from the second row of (Equation 8), the following identical equation is expressed:(9)$$\begin{array}{c} R_{s}\left(i_{bF}-i_{cF}\right)-\left(L-M\right)\frac{d(i_{bF}-i_{cF})}{dt}=0, \end{array}$$
regardless of $R_{s}$, *L* and *M*, the only solution that satisfies (Equation 9) is:(10)$$\begin{array}{c} i_{bF}=i_{cF}. \end{array}$$

By (Equation 6) and (Equation 10), the relation of faulty currents is:(11)$$\begin{array}{c} i_{aF}=-2i_{bF}=-2i_{cF}. \end{array}$$

Similarly, subtracting the second row from the first row of (Equation 8) and then the relationship of (Equation 11), the following identical equation is expressed:(12)$$\begin{array}{c} R_{s}\left(\frac{3}{2}i_{aF}-\mu i_{f}\right)-\left(L-M\right)\frac{d(\frac{3}{2}i_{aF}-\mu i_{f})}{dt}=0, \end{array}$$
regardless of $R_{s}$, *L* and *M*, the only solution that satisfies (Equation 12) is:(13)$$\begin{array}{c} \frac{3}{2}i_{aF}=\mu i_{f}. \end{array}$$

Using (Equation 2), the current of phase *a* of the faulty part is:(14)$$\begin{array}{c} i_{aF}=\frac{2}{3}\frac{\mu^{2}v_{a}}{R_{f}+\mu \left(1-\mu \right)R_{s}}=G\left(\mu ,R_{f}\right)v_{a}, \end{array}$$
where $G(\mu ,R_{f})$ is the factor associated with the ITSF. $G(\mu ,R_{f})$ is zero when normal, and increases when μ increases or $R_{f}$ decreases.

In a similar manner, when ITSF occurs on phase *b* and *c*, the current of phase *b* and *c* of the faulty part are as follows, respectively:(15)$$\begin{array}{c} i_{bF}=\frac{2}{3}\frac{\mu^{2}v_{b}}{R_{f}+\mu \left(1-\mu \right)R_{s}}=G\left(\mu ,R_{f}\right)v_{b}, \end{array}$$
(16)$$\begin{array}{c} i_{cF}=\frac{2}{3}\frac{\mu^{2}v_{b}}{R_{f}+\mu \left(1-\mu \right)R_{s}}=G\left(\mu ,R_{f}\right)v_{c}. \end{array}$$

### 3.2. Analysis of PSC and NSC with ITSF

Symmetrical components are one of the significant indicators of unbalance in three-phase systems [19]. Among the symmetrical components, positive- and negative-sequence components for the three-phase currents are expressed by following equations:(17)$$\begin{array}{c} i_{PS}=\frac{1}{3}\left(i_{a}+e^{2\pi j/3}\cdot i_{b}+e^{4\pi j/3}\cdot i_{c}\right), \end{array}$$
(18)$$\begin{array}{c} i_{NS}=\frac{1}{3}\left(i_{a}+e^{4\pi j/3}\cdot i_{b}+e^{2\pi j/3}\cdot i_{c}\right), \end{array}$$
where $i_{PS}$ and $i_{NS}$ are positive- and negative-sequence current, respectively.

Assuming that ITSF has occurred in phase *a*, using (Equation 4), (Equation 5) and (Equation 14), PSC is expressed as:(19)$$\begin{array}{c} i_{PS}=i_{aH}+\frac{1}{2}G\left(\mu ,R_{f}\right)v_{a}. \end{array}$$

Similarly, using (Equation 4), (Equation 5) and (Equation 14), NSC is expressed as:(20)$$\begin{array}{c} i_{NS}=\frac{1}{2}G\left(\mu ,R_{f}\right)v_{a}. \end{array}$$

For in-depth analysis, (Equation 19) and (Equation 20) can be divided into magnitude and phase parts, respectively. The magnitudes of PSC and NSC are expressed as follows, respectively:(21)$$\begin{array}{c} |i_{PS}|=|i_{aH}+\frac{1}{2}G\left(\mu ,R_{f}\right)v_{a}|, \end{array}$$
(22)$$\begin{array}{c} |i_{NS}|=\frac{1}{2}G\left(\mu ,R_{f}\right)|v_{a}|. \end{array}$$

In (Equation 21), the magnitude of PSC increases as ITSF increases. The magnitude of NSC is proportional to $G(\mu ,R_{f})$ that means the degree of ITSFs in (Equation 22). Comparing (Equation 21) with (Equation 22), since (Equation 21) has term $i_{aH}$, (Equation 22) stands out more directly in ITSF change. Therefore, we determine (Equation 22) as a magnitude indicator.
(23)$$\begin{array}{c} Magnitude\;Indicator=\frac{1}{2}G\left(\mu ,R_{f}\right)\left|v_{a}\right|. \end{array}$$

The magnitude indicator remains at zero in a normal state, while it increases when ITSF increases. In particular, since the magnitude indicator depends on the magnitude of the current, it is advantageous for ITSF diagnosis as the torque increases. On the other hand, there is also a small amount of the magnitude indicator since currents are small when torque is low. Therefore, under no-load or low torque, the magnitude indicator may be difficult to diagnose even if a slight ITSF occurs.

Meanwhile, in the case of the phase part, we pay attention is the phase difference between the two rather than the characteristics of each PSC and NSC. Using (Equation 19) and (Equation 20), the phase indicator is defined as follows:(24)$$\begin{array}{c} Phase\;Indicator=∠\left(i_{aH}+\frac{1}{2}G\left(\mu ,R_{f}\right)v_{a}\right)-∠\left(\frac{1}{2}G\left(\mu ,R_{f}\right)v_{a}\right). \end{array}$$

Figure 2 represents vector plots of the phase indicator for different two ITSFs under the same operating conditions. The left of Figure 2 means the angle between the vector $\frac{1}{2}G\left(\mu ,R_{f}\right)v_{a}$ and vector $i_{aH}+\frac{1}{2}G\left(\mu ,R_{f}\right)v_{a}$ when the degree of ITSFs is $G(\mu ,R_{f})$. The right of Figure 2 shows when the degree of ITSFs is $G^{\prime }\left(\mu ,R_{f}\right)$. In the right of Figure 2, the phase indicator also means the angle between the vector $\frac{1}{2}G^{\prime }\left(\mu ,R_{f}\right)v_{a}$ and vector $i_{aH}+\frac{1}{2}G^{\prime }\left(\mu ,R_{f}\right)v_{a}$ when the degree of ITSFs is $G^{\prime }\left(\mu ,R_{f}\right)$. Figure 2 represents the degree of ITSFs increases from *G* to $G^{\prime }$, the phase indicator decreases. Therefore, the phase indicator has a characteristic that decreases as the ITSF increases. In addition, since the phase indicator depends on the phase difference, it may be advantageous for weak ITSF diagnosis compared to the magnitude indicator even at an operating point with a low current.

### 3.3. Proposed Fault Indicator

In order to diagnose the slight ITSF as well as the degree of ITSF under various operating conditions, we propose the following fault indicator by combining (Equation 23) and (Equation 24).
(25)$$\begin{array}{c} FI=|i_{NS}|\cos\left(∠i_{PS}-∠i_{NS}\right). \end{array}$$

The phase part of (Equation 25) is set similarly to the definition of power factor angle. The proposed fault indicator has a characteristic that increases as ITSF increases. In addition, a slight ITSF can be diagnosed by the effect of the phase indicator in a low current condition, and the magnitude indicator in a high current condition.

Figure 3 illustrates the proposed method in a flow chart. The proposed method using only three-phase currents can diagnose the degree of the slight ITSF and the degree of ITSFs under various operating conditions.

## 4. Experimental Setup and Data Collection

PMSM operation system in Figure 4 was set up to investigate the performance of the proposed fault indicator. PMSM was controlled by the Inverter-Starvert-iG5A manufactured by LSIS and coupled with a dynamometer to generate and measure the load torque in real time. In addition, the PMSM used in the experiment is FMAIN22-BBFB2 PMSM manufactured by Higen, and its specifications are given in Table 1. The motor is a three phase 220 Vac PMSM with 6 poles and 36 slots. The current signals were measured using three ammeters, and the rotational angle was measured using an encoder. The measurement values were collected using LabVIEW with cDAQ-9178 at the sampling frequency of 100 KHz. Moreover, the phase indicator was calculated using the difference between the phase of the amplitude of the PSC waveform and the phase of the amplitude of the NSC waveform.

In order to confirm the performance of the proposed fault indicator, experiments were conducted at torques of no-load (Figure 5a), 1, 2, 3, 4, and 4.5 Nm (Figure 5b) and rotational speeds of 2000, 3000, 4000, and 4500 rpm (Figure 6) using under normal and several fault PMSMs.

Table 2 and Table 3 show the ITSF conditions according to μ and $R_{f}$, respectively, and ITSFs that are their combinations of Table 2 and Table 3 were tested in this experiment. In addition, to make ITSF, a short circuit was made on the winding and the fault resistance was connected by the resistance. In this experiment, the slightest ITSF is when μ=0.042 and $R_{f}=2.128\;\Omega$, and the most severe ITSF is when μ=0.125 and $R_{f}=0.218\;\Omega$.

Figure 7a represents the time-domain waveforms of the phase current and fault current at slightest ITSF when μ is 0.042 and $R_{f}$ is 2.128Ω at 4500 rpm. Figure 7a shows that the magnitude of the fault current is much smaller than the magnitude of the phase current. Figure 7b represents the amplitude values of the fault current with rotor speed at the slightest ITSF, and it is shown that the amplitude of the fault current is less than 3 A at all rotor speeds. Therefore, compared to existing methods [22,23], the slight ITSF conditions were well set up in this experiment.

## 5. Experimental Results and Discussion

### 5.1. PSC and NSC

To confirm the characteristics of PSC and NSC with ITSF, Figure 8 and Figure 9 represent the time-domain waveforms of PSC and NSC with μ and $R_{f}$, respectively. Since features in ITSF become more evident in severe fault conditions, three groups were chosen in this experiment: one normal and two severe ITSFs with μ=0.082 and μ=0.125 under fixed minimum $R_{f}=0.218\;\Omega$ in Figure 8. PSCs are minimaly changed in the amplitude with μ, whereas the phase shift is easily observed in Figure 8a. For NSCs, the amplitude gradually increases and the phase also changes as μ increases in Figure 8b.

For the same reason as with aforementioned experiment, we confirm the time-domain waveforms of PSC and NSC with $R_{f}$ of 0.366Ω and 0.218Ω under fixed maximum μ=0.125 in Figure 9. It can be observed that the amplitude of PSC with $R_{f}$ variations is not distinguished in the fault cases from the normal case, whereas the phase shifts of PSC are represented with $R_{f}$ variations in Figure 9a. In the case of NSC, as $R_{f}$ gradually decreases, the amplitude of NSC gradually increases, and the phase shifts of NSC make a distinction between the normal case and the two fault cases in Figure 9b.

These results mean that PSC and NSC differ in ways. While the phase of PSC is affected by ITSF, NSC shows both a phase shift and an increase in amplitude. Such a result is in clear agreement with the results discussed in Section 3.

### 5.2. Vector Diagrams for Magnitude and Phase Indicators

Figure 10 illustrates the vector diagrams of the magnitude and phase indicators under various speeds and no-load. Each vectors indicate the normal and three ITSF cases with μ, and the fault condition is composed of μ1=0.042, μ2=0.083, and μ3=0.125 for fixed minimum $R_{f}=2.128\;\Omega$. Figure 10a shows that the vector changes clockwise when it changes from normal conditions to increasing μ conditions, which confirms that the phase indicator decreases as the ITSF increases. In addition, the slightest ITSF, the green-colored vector, and the normal, the black-colored vector, are well distinguished, and the degree of ITSF is also diagnosed. On the other hand, the magnitude of each vector excepting when μ is 0.125 varies little, indicating difficulty in diagnosing ITSF with only the magnitude indicator when the torque is no-load.

Similarly, Figure 10b–d represent that difficult to diagnose ITSF with only the magnitude indicator under no-load, but it is confirmed that the slightest ITSF and the degree of ITSF are diagnosed with the phase indicator. Therefore, ITSF can be diagnosed under no-load only when the proposed fault indicator is used in consideration of both magnitude and phase indicators.

Figure 11 describes the vector diagrams of the magnitude and phase indicators with $R_{f}$ when the operating condition is no-load and various speeds. The fault condition is composed of $R_{f}1=2.128\;\Omega$, $R_{f}2=0.366\;\Omega$, and $R_{f}3=0.218\;\Omega$ for fixed maximum μ=0.042. Each vector indicates the normal and three ITSF cases with $R_{f}$. The vectors in Figure 11a change clockwise with $R_{f}$, which confirms that the phase indicator decreases as $R_{f}$ decreases. The vectors in Figure 11 indicates that the vectors change clockwise with $R_{f}$, and the phase indicator decreases as the ITSF increases. In addition, the slightest ITSF, the green-colored vector, and the normal, the black-colored vector, are well distinguished, and the degree of ITSF is also diagnosed. For the magnitude of each vector, the normal and slightest ITSF are indistinguishable, and the blue-colored and red-colored vectors slightly increased as the $R_{f}$ decreased. This means that it is difficult to diagnose slight ITSF with only the magnitude indicator under no-load.

Figure 12 depicts the vector diagrams by the combination of the magnitude and phase indicators under 4.5Nm and varying speeds. The fault condition is composed of μ1 = 0.042, μ2 = 0.083 and μ3 = 0.125 for fixed maximum $R_{f}=2.128\;\Omega$. The black-colored vector in Figure 12a represents the normal case, and the other vectors indicate the ITSF cases. The magnitude of the vectors in Figure 12a gradually increases as μ increases. In addition, the slightest ITSF, the green-colored vector, and the normal, the black-colored vector, are well distinguished, and the degree of ITSF is also diagnosed. On the other hand, for the phase of vectors, the normal and slightest ITSF are indistinguishable, and the blue-colored and red-colored vectors slightly change clockwise as μ increases.

Figure 12b–d show similar trends as in Figure 12a. The phase indicator is difficult to diagnose ITSF under high torque, whereas the magnitude indicator diagnoses the slightest ITSF and the degree of ITSF. This means that the magnitude indicator is easy to diagnose the slight ITSF and the degree of ITSF. Therefore, ITSF diagnosis under high torque greatly needs the proposed fault indicator by consideration of both magnitude and phase indicators.

Figure 13 represents the vector diagrams of the magnitude and phase indicator under 4.5Nm and varying speeds. The fault condition is composed of $R_{f}1=2.128\;\Omega$, $R_{f}2=0.366\;\Omega$, and $R_{f}3=0.218\;\Omega$ for fixed minimum μ=0.042. The phase of the vector in Figure 13a fails to distinguish the normal and ITSF cases. On the other hand, the magnitude of vector gradually increases as $R_{f}$ decreases in Figure 13a. It shows that the magnitude indicator acts as the fault indicator under high torque.

Meanwhile, Figure 13b–d describes that the phase of vector does not distinguish between normal and the slightest ITSF and can distinguish other ITSFs. On the other hand, the magnitude of vector gradually increases as $R_{f}$ increases. It means that ITSFs can be diagnosed by combination of the magnitude and the phase.

Figure 10, Figure 11, Figure 12 and Figure 13 show the magnitude and phase indicators under various fault conditions and operating conditions. The results explain why both the magnitude of the phase information of the currents is needed to diagnose ITSF under various conditions.

### 5.3. Fault Indicator under Various Conditions

To comfirm the performance of the proposed fault indicator, Figure 14 and Figure 15 represent the several points under various torque and speeds. The black, green, blue, and red points of Figure 14 and Figure 15 decscribe the fault indicator under 2000, 3000, 4000, and 4500 rpm, respectively.

Figure 14 represents the fault indicator with μ, and the fault condition is composed of μ1 = 0.042, μ2 = 0.083, and μ3 = 0.125 for fixed maximum $R_{f}=2.128\;\Omega$. The red points in Figure 14a show the fault indicator with μ at 4500 rpm, which is rated speed. When μ is zero, the fault indicator has the smallest value, and it increases as μ increases. Similarly, at 2000, 3000, and 4000 rpm, the fault indicators are confirmed that gradually increase as μ increases, respectively. In particular, the black points at 2000 rpm distinguish from normal and the slightest ITSF, μ = 0.042. In addition, the proposed fault indicator shows that the degree of ITSF also increases gradually as μ increases.

Similarly, even when there is a load, the proposed fault indicators increase as μ increases as shown in Figure 14b–f. Additionally, in Figure 14, the fault indicator has a trend to increase as the speed increases or the torque increases under the same ITSF condition.

Figure 15 describes the fault indicator for normal and various ITSF conditions with inversed value of $R_{f}$ under no-load. The fault condition is composed of $R_{f}1=2.128\;\Omega$, $R_{f}2=0.366\;\Omega$, and $R_{f}3=0.218\;\Omega$ for fixed minimum μ=0.042. The fault indicator in Figure 15a increases as the inversed value of $R_{f}$ increases. In particular, the proposed fault indicator distinguishes the slightest ITSF from normal even at 2000 rpm, an operating point that is difficult to diagnose, and diagnoses the degree of ITSF.

Similarly, even under load in Figure 15b–f, the proposed fault indicator gradually increases as the inversed value of $R_{f}$ increases at the same operating points. In addition, it is confirmed that for all operating points, the slightest ITSF, μ = 0.042, and normal are distinguished. In conclusion, since the proposed fault indicator considers both magnitude and phase information, only measered three-phase currents diagnose the slight ITSF as well as the degree of ITSF at various torque and speeds.

## 6. Conclusions

In this paper, we propose a simple and effective method of detecting slight ITSF and severity of ITSFs by defining a novel fault indicator based on the magnitude and phase of electric current. The fault indicator was proposed based on an analysis of PSC and NSC with ITSFs. Our proposed method uses only the three-phase electric current and diagnose ITSFs before they become serious without information regarding the machine parameters, voltage, or rotational angles. The experiment results are demonstrated that the suggested fault indicator diagnoses the slight and degree of ITSFs depending on the shorted-turn ratio or fault resistance, in addition to confirming the performance of the proposed fault indicator under various torque and speeds. The suggested fault indicator is useful in industrial environments wherein sensorless control machines or aged machines render it impossible to identify the changed parameter values.

## Figures and Tables

**Figure 1 sensors-22-04597-f001:**
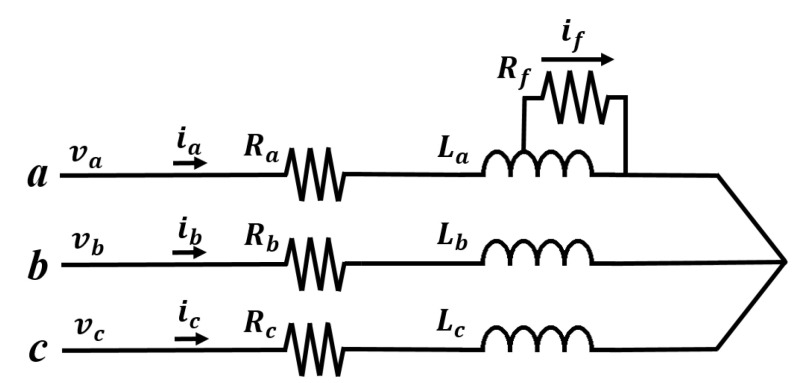
Equivalent model of a PMSM with an ITSF in the phase *a*.

**Figure 2 sensors-22-04597-f002:**
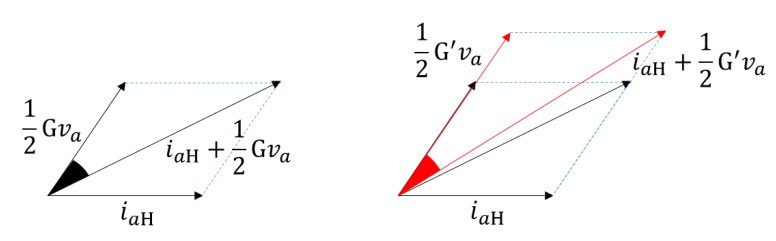
Vector plots of the degrees of two ITSFs under the same operating conditions.

**Figure 3 sensors-22-04597-f003:**
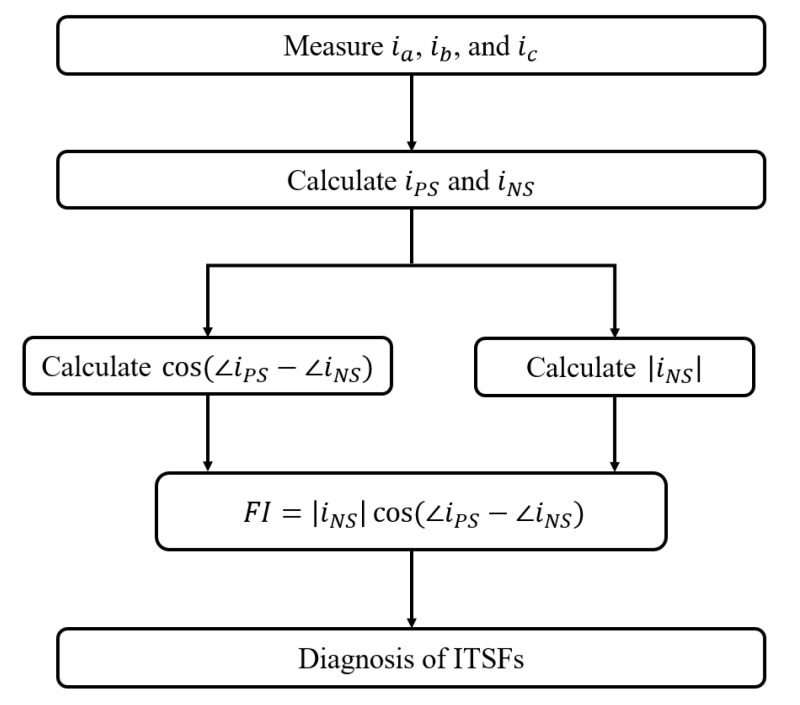
Flow chart for diagnosis of ITSFs in a PMSM.

**Figure 4 sensors-22-04597-f004:**
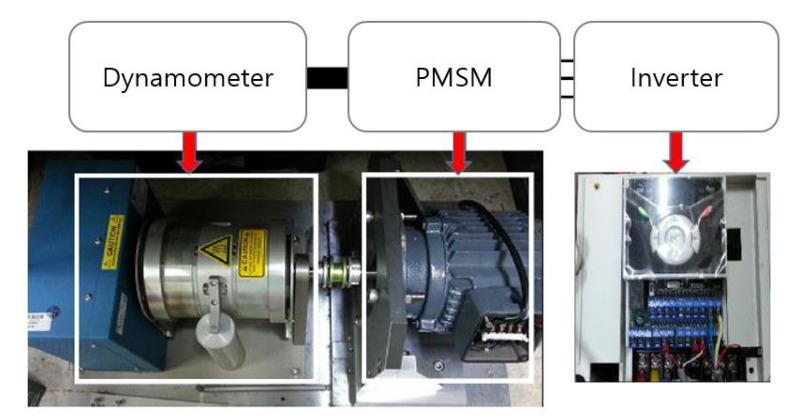
Experimental setup.

**Figure 5 sensors-22-04597-f005:**
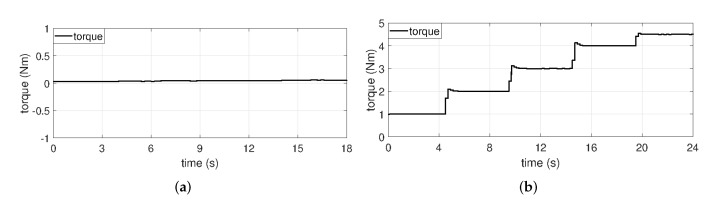
Torque at (**a**) no-load and (**b**) load.

**Figure 6 sensors-22-04597-f006:**
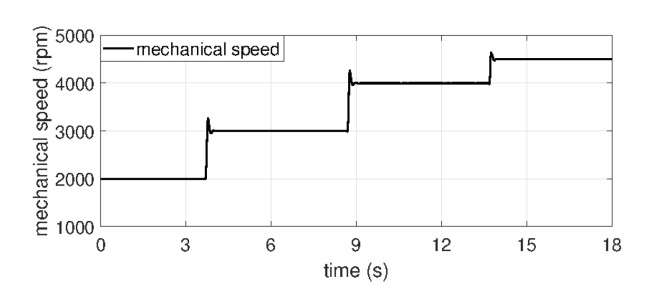
Mechaincal rotor speed.

**Figure 7 sensors-22-04597-f007:**
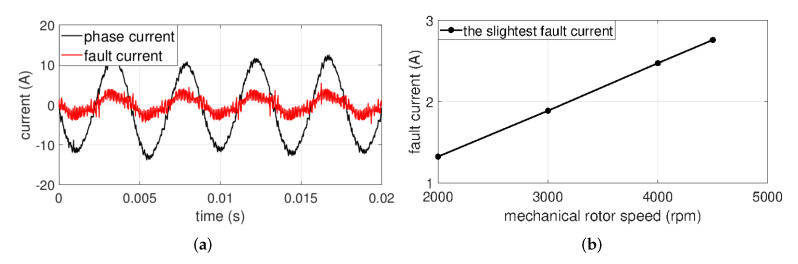
(**a**) Phase current and fault current at rated conditions and (**b**) the slightest fault current with mechanical rotor speed.

**Figure 8 sensors-22-04597-f008:**
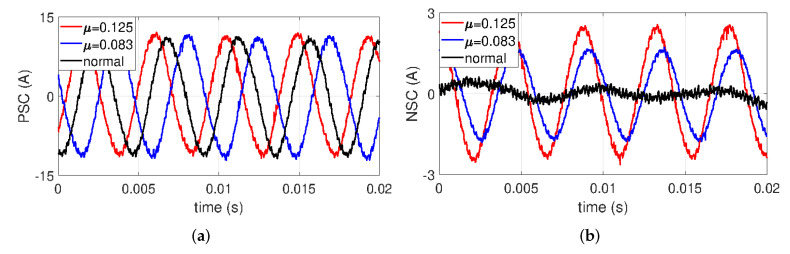
(**a**) PSC and (**b**) NSC with μ at rated conditions.

**Figure 9 sensors-22-04597-f009:**
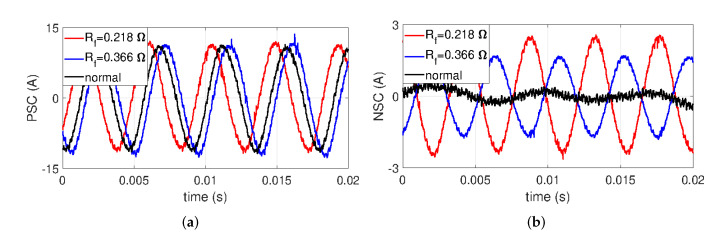
(**a**) PSC and (**b**) NSC with $R_{f}$ at rated conditions.

**Figure 10 sensors-22-04597-f010:**
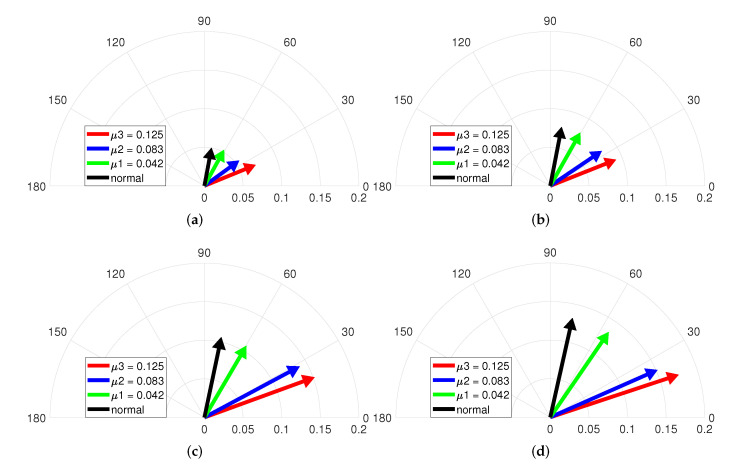
Vector diagram at (**a**) 2000, (**b**) 3000, (**c**) 4000, and (**d**) 4500 rpm and no-load with μ.

**Figure 11 sensors-22-04597-f011:**
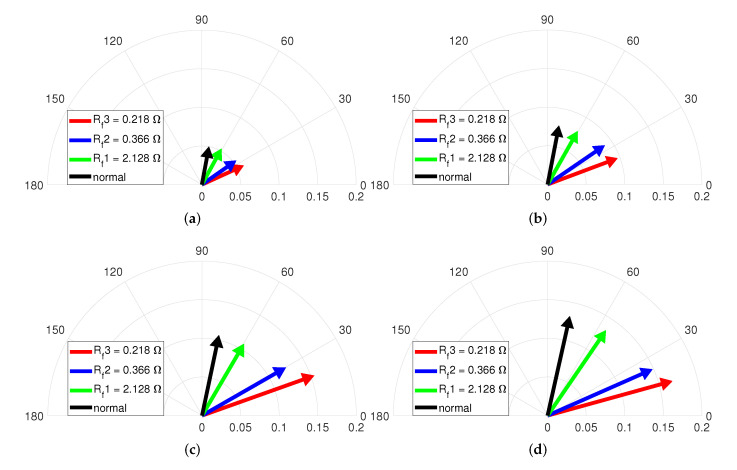
Vector diagram at (**a**) 2000, (**b**) 3000, (**c**) 4000, and (**d**) 4500 rpm and no-load with $R_{f}$.

**Figure 12 sensors-22-04597-f012:**
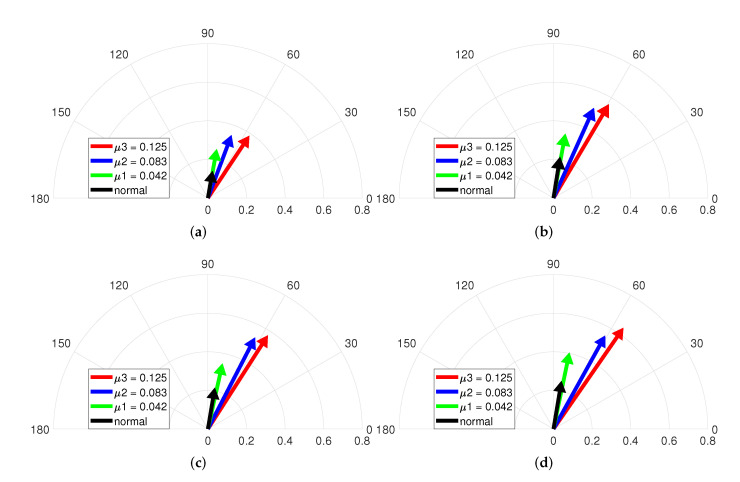
Vector diagram at (**a**) 2000, (**b**) 3000, (**c**) 4000, and (**d**) 4500 rpm and 4.5 Nm with μ.

**Figure 13 sensors-22-04597-f013:**
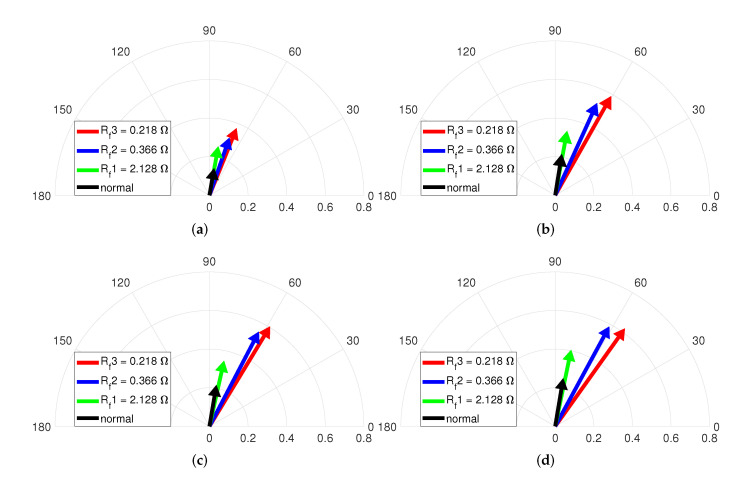
Vector diagram at (**a**) 2000, (**b**) 3000, (**c**) 4000, and (**d**) 4500 rpm and 4.5 Nm with $R_{f}$.

**Figure 14 sensors-22-04597-f014:**
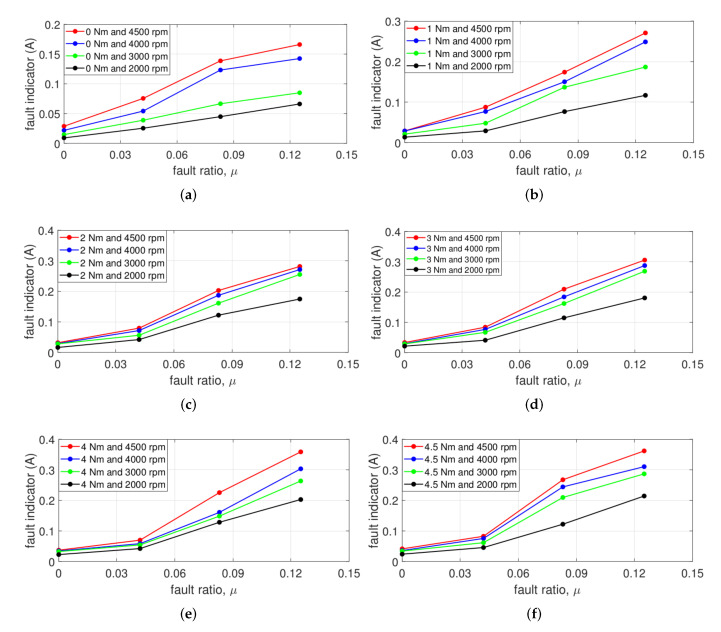
Fault indicator with μ at (**a**) 0, (**b**) 1, (**c**) 2, (**d**) 3, (**e**) 4, and (**f**) 4.5 Nm.

**Figure 15 sensors-22-04597-f015:**
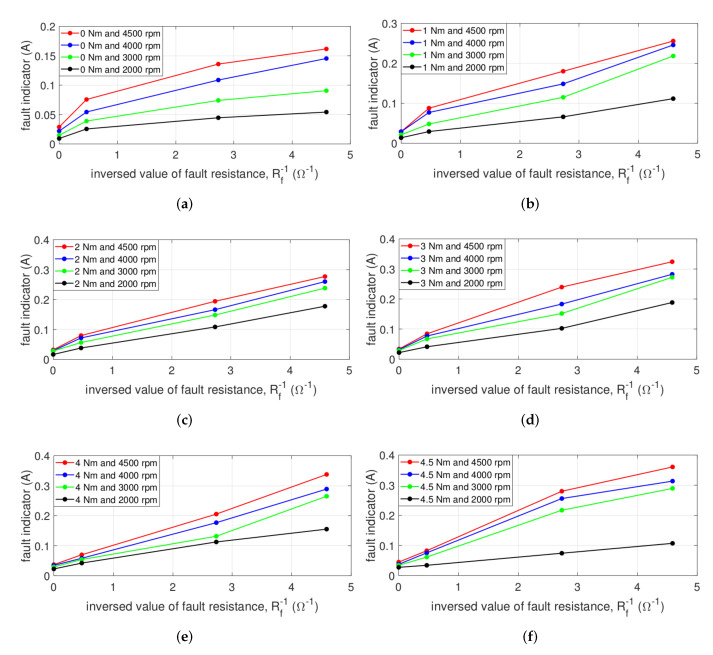
Fault indicator with the inversed value of $R_{f}$ at (**a**) 0, (**b**) 1, (**c**) 2, (**d**) 3, (**e**) 4, and (**f**) 4.5 Nm.

**Table 1 sensors-22-04597-t001:** Specifications of PMSM used in the experiments.

Parameters	Values	Units
Rated output	2.2	kW
Rated speed	4500	rpm
Rated torque	4.5	Nm
Rated current	10.2	A
Input voltage	220	V
Number of pole pairs	3	-
Self inductance, $L_{ms}$	2.42	mH
Leakage inductance, $L_{ls}$	0.67	mH
Stator resistance, $R_{s}$	0.42	Ω

**Table 2 sensors-22-04597-t002:** Used μ in the experiments.

μ1	μ2	μ3
0.042	0.083	0.125

**Table 3 sensors-22-04597-t003:** Used $R_{f}$ in the experiments.

$R_{f}1$	$R_{f}2$	$R_{f}3$
2.138 Ω	0.366 Ω	0.218 Ω

## Data Availability

Not applicable.

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
