# Peer review of "Interturn Short Fault Diagnosis Using Magnitude and Phase of Currents in Permanent Magnet Synchronous Machines"

_sensors, 2022, doi:10.3390/s22124597_

Round 1
Reviewer 1 Report
the manuscript has a problem: how to collect current and phase? wish author detally explain test process. otherwise I feel your data is simulation.
Author Response
Thank you for your insightful comment.
We carefully revised it reflecting your comments.
We deeply appreciate your comments.
We attached the pdf file for the specific response.
We wish your affirmative decision on our revision.

Reviewer 2 Report
sensors-1689520
Slight Inter-turn Short Fault Diagnosis Only Based on Magnitude and Phase of Currents in Permanent Magnet Synchronous Machines
A fault indicator using magnitude and phase of currents is developed for detecting inter-turn short faults in permanent magnet synchronous machines. The fault indicator has been checked by numerical and experimental investigations. Generally, the work is good enough to be published. However, we have following questions:
1. Title should be revised and highlighted the main work you done. For example, ‘Inter-turn Short Fault Diagnosis Using Magnitude and Phase of Currents in Permanent Magnet Synchronous Machines’.
2. From the Abstract, it seems that ‘a novel fault indicator’ should be given in details.
3. For the ‘inter-turn short fault’, how to define ‘slight’?
4. If possible, could you kindly give more details about the influence of environmental noises?
Author Response
We deeply appreciate the reviewer's insightful comments that helped us improve the quality of our manuscript.
We carefully studied all your comments, and revised the manuscript.
We wish your affirmative decision on our revision.

Round 2
Reviewer 1 Report
this version is OK, I have not any comments